

# Characteristics and usefulness of trunk muscle endurance tests on the Roman chair in healthy adults

Maja Petrič[1], Lijana Zaletel-Kragelj[2] and Renata Vauhnik[1]

[1] Department of Physiotherapy/Faculty of Health Sciences, University of Ljubljana, Ljubljana, Slovenia
[2] Department of Public Health/Faculty of Medicine, University of Ljubljana, Ljubljana, Slovenia

Corresponding author
Renata Vauhnik,
renata.vauhnik@zf.uni-lj.si

## ABSTRACT

**Background:** Adequate trunk muscle endurance is considered to be an important indicator of good low back stability; therefore, its assessment is needed when determining an individual's risk for back pain. Optimal tests to assess each trunk muscle group separately are difficult to find. The objective of this study was to verify if two groups of trunk muscle endurance tests (standard and alternative) show comparable results in terms of muscle endurance ratios, holding times and rated perceived effort to perform each test.

**Methods:** The study was designed as a quasi-experimental repeated-measures design. There was a single group of participants who took part in two different trunk muscle endurance testing. Sixty-eight healthy adult volunteers, aged 20–45 years (31.9 ± 7.2 years), without recent musculoskeletal injury or disorder participated in the study. All participants finished the study. Trunk muscle endurance tests as tested on the Roman chair (B tests) were compared with standard tests as suggested by McGill (A tests). Each group of tests consisted of an endurance test for trunk extensors, trunk flexors, and lateral trunk muscles for left and right side. The order of tests' performances was randomly assigned to each participant, whereby a participant did perform A and B tests in the same order. In each test of A and B the holding time was recorded and a perceived effort in each test performance was also assessed by participants. Post testing performance the four ratios of trunk muscles endurance comparison were calculated for each group of tests to determine if there is a good or poor ratio between muscles. Results of each participant were compared for trunk muscle endurance ratio calculations, holding times and rated perceived effort for A and B tests.

**Results:** Results showed comparable trunk muscle endurance ratios in the three ratios observed, except for the flexors:extensors ratio ($A_{FL:EX}$: 1.2 (IQR: 0.7–1.6) *vs.* $B_{FL:EX}$: 0.6 (IQR: 0.3–0.8); $p < 0.001$). As compared to A tests, holding times were significantly longer in B tests for the extensors ($A_{EX}$: 125.5 s (IQR: 104.8–182.8 s) *vs.* $B_{EX}$: 284.0 s (IQR: 213.0–342.3 s); $p < 0.001$) and lateral trunk muscles ($A_{L-LM}$: 61.0 s (IQR: 48.3–80.8 s) *vs.* $B_{L-LM}$: 131.5 s (IQR: 95.5–158.5 s); $A_{R-LM}$: 63.5 s (IQR: 45.8–77.3 s) *vs.* $B_{R-LM}$: 113.0 s (IQR: 86.3–148.8 s); $p < 0.001$), both were also rated as slightly easier to perform in the extensors ($A_{RPE-EX}$: 13 (IQR: 12.0–14.0) vs $B_{RPE-EX}$: 11 (IQR: 10.0–13.0); $p_{RPE-EX} < 0.001$) and lateral muscles testing ($A_{RPE-LM}$: 14.0 (IQR: 12.3–15.8) *vs.* $B_{RPE-LM}$: 13.0 (IQR: 12.0–15.0); $p_{RPE-LM} = 0.001$).

**Conclusions:** A and B tests are comparable in three of four trunk muscle endurance ratios, while longer holding times and lower perceived effort to perform were observed in most of the B tests. The Roman chair tests could be used as an alternative to standard tests.

## INTRODUCTION

Musculoskeletal disorders are an important global public health problem, representing almost one fifth of all non-communicable diseases worldwide, as well as in Slovenia (15.6% of the total burden in terms of the Years Lived with Disability measure (YLDs)) (*Institute for Health Metrics & Evaluation (IHME), 2022*). Among these disorders and COMA low back pain (LBP) was the leading cause of functional disability in terms of the YLDs (*Chen et al., 2021*). LBP is present in all age groups and the prevalence, incidence and YLDs rates are higher in women (especially in the middle aged women) than in men and it increases with age for both genders (*Hartvigsen et al., 2018*; *Chen et al., 2021*; *Institute for Health Metrics & Evaluation (IHME), 2022*). In the majority of patients, a specific pathological cause of LBP remains undefined, but there are some lifestyle factors (such as insufficient physical activity, sedentary lifestyle and thereby poor physical performance) associated with the occurrence of LBP (*Hartvigsen et al., 2018*; *Mahdavi et al., 2021*; *Dzakpasu et al., 2021*).

In addition to spine movement, the primary function of the major trunk muscle groups (extensors, flexors and the lateral trunk muscles) are to provide stability and active support of the spine and pelvis during virtually every movement of the body. Therefore, inadequate or inappropriate trunk muscle performance or inappropriate muscle endurance ratios among the major trunk muscle groups are considered to be important risk factors for LBP occurrence as well as risk factors for other musculoskeletal disorders by inference (*Panjabi, 1992*; *McGill, Childs & Liebenson, 1999*; *McGill, 2016*).

As low back stability depends on the endurance of all four above-mentioned trunk muscle groups, assessment of the endurance of each trunk muscle group individually and their endurance ratio calculation are needed when determining an individual's risk for spine health problems (*McGill, 2016*). According to some authors (*Biering-Sørensen, 1984*; *McGill, Childs & Liebenson, 1999*; *Martínez-Romero et al., 2020*; *Tavares et al., 2020*), the endurance ratios of major trunk muscle groups differ significantly between individuals with LBP and those without a history of LBP. Therefore, feasible trunk muscle endurance assessment could present an effective evidence-based public health measure for prevention and treatment in musculoskeletal rehabilitation of lumbar spine conditions. A proposed battery of tests could be very useful in continuous monitoring of the lower back musculature in children, adolescents, and emerging adults, especially students. In these population groups, according to the recent research, which showed alarming results (*Jurak et al., 2021*; *Tišlar, Starc & Kukec, 2022*), such monitoring will be more than necessary.
Optimal endurance tests which isolate the trunk extensors, flexors, and lateral trunk muscle groups are difficult to find (*McGill, 2016*). One of the most reliable and frequently used endurance tests are tests proposed by McGill (*McGill, Childs & Liebenson, 1999*; *McGill, 2016*). However, many authors have emphasized that the endurance of other muscles such as hip and shoulder muscles can contribute to the performance of above mentioned endurance tests (*Pagé, Dubois & Descarreaux, 2011*; *McGill, 2016*; *Tuff, Beach & Howarth, 2020*; *Castro-Piñero et al., 2021*; *Juan-Recio et al., 2022*). Therefore, some of those testing positions are not suitable for some groups of participants (*e.g.*, older adults, individuals with upper limb injuries, pain or weakness *etc.*) (*Ledoux, Dubois & Descarreaux, 2012*; *Pagé & Descarreaux, 2012*). Furthermore, the body position angle of inclination, as well as the type of the kinetic chain (open *vs.* closed) differs between the tests and therefore can affect the muscle activation and holding times (*Tuff, Beach & Howarth, 2020*).

As currently available and used tests have some limitations in the use of different population groups as well as in the positions used for testing, there is a need for optimizing the trunk muscle endurance assessment protocols. One, potentially better, alternative protocol when testing trunk muscle endurance is testing it on the 45° Roman chair apparatus (*Ledoux, Dubois & Descarreaux, 2012*; *Pagé & Descarreaux, 2012*). The Roman chair is mainly used as an exercise tool rather than as testing equipment for the back muscles (*Shigaki et al., 2018*). The difference in low back loading demands of the open- and closed-chain variations of some tests on the Roman chair were already studied. Authors wanted to determine if the recommended alternative 45° inclination angle for the open-chain testing position (when testing on the Roman chair) are a suitable approximation of the low back loading demands of the closed-chain testing position (when testing with tests proposed by McGill) (*Tuff, Beach & Howarth, 2020*). For the lateral trunk muscles *Tuff, Beach & Howarth (2020)* obtained 45° inclination angle for the open-chain side bridge provides a suitable approximation of the low back lateral bend reaction moment in the closed-chain side bridge.

Suitability of the trunk muscle endurance assessment on the Roman chair in various population groups has not been studied yet. Although some of the trunk muscle endurance tests on the Roman chair have been described in a few studies (for extensors and lateral trunk muscles endurance testing), the characteristics and usefulness of the proposed group of tests (*i.e.*, for each of three major trunk muscle groups) have not been studied in detail.

The aim of this study was to provide the evidence for effective public health measure in the trunk muscle endurance assessment suitable for various population groups (regardless of age, musculoskeletal problems, weakness *etc.*). Therefore, our objective was to study the characteristics and usefulness of potentially better alternative test positions when measuring trunk muscle endurance and to verify if two different groups of trunk muscle endurance tests shows comparable results in terms of muscle endurance ratio calculations, holding times and rated perceived effort to perform each test. We hypothesized that tests groups will show comparable results in terms of the muscle endurance ratios in healthy adults, with tests on the Roman chair having longer holding times; and being easier to perform.

## MATERIALS AND METHODS

### Study design

The study was designed as a quasi-experimental repeated-measures design (*Rogers & Révész, 2019*). There was a single group of participants who took part in two different trunk muscle endurance testing as described below in the Study procedures section.

### Study sample

Ninety potential candidates have been invited initially to voluntarily participate in our study. They were invited by using electronic media (email, Facebook, *etc.*) and chain referral sampling. Of those, eighty-two were willing to cooperate and were screened for inclusion criteria: (a) healthy adults aged between 20–45 years old, (b) without LBP in time of inclusion into the study, and (c) without musculoskeletal injures or other illness, that could represent contraindication for the muscle endurance testing or pose a risk to an individual's health. Fourteen of them have not fulfilled all the inclusion criteria and were consequently excluded.

Therefore, sixty-eight healthy adults (51 women (75%) and 17 men (25%); age: $31.9 \pm 7.2$ years; body height: $1.7 \pm 0.1$ m; body mass: $66.4 \pm 12.1$ kg; body mass index: $23.0 \pm 3.5$ kg/m$^2$), who signed an informed consent, participated in the study and all participants finished the study.

### Study instruments

#### Questionnaires

All participants completed questionnaires on their demographic data, health status and physical activity (*Jakovljević, Knific & Petrič, 2017*). Questions on health status and physical activity were intended to reveal if some exclusion criteria or contraindication exists among the candidates to participate in the study.

#### Trunk muscle endurance tests

The endurance of four major trunk muscle groups (trunk extensors (EX), trunk flexors (FL) and lateral trunk muscles (LM)–right (R-LM) and left side (L-LM)) was measured. The endurance tests performed in this study are referred to as "A tests" (Fig. 1), and "B tests" (Fig. 2).

A tests, following protocols already established by *McGill, Childs & Liebenson (1999)*, consisted of the trunk extensor endurance test (known also as the Biering-Sørensen test (*Biering-Sørensen, 1984*); intra-class correlation coefficient (ICC) (95% confidence interval (CI)): 0.94 [0.84–0.98]) (*Martínez-Romero et al., 2020*), the trunk flexor endurance test (ICCs: 0.93–0.95) (*Cuenca-Garcia et al., 2022*), and the side bridge test on both right and left sides (ICC (95% CI): 0.81 [0.60–0.91]) (*Juan-Recio et al., 2022*) (Fig. 1):

- The trunk extensor endurance test (Test EX-A) was performed with participants lying prone on a treatment table, positioning the anterior superior iliac spines (lat. spina iliaca anterior superior—SIAS) at the table edge while supporting the upper body with arms placed on a chair. For lower body stabilization their pelvis and lower legs were anchored

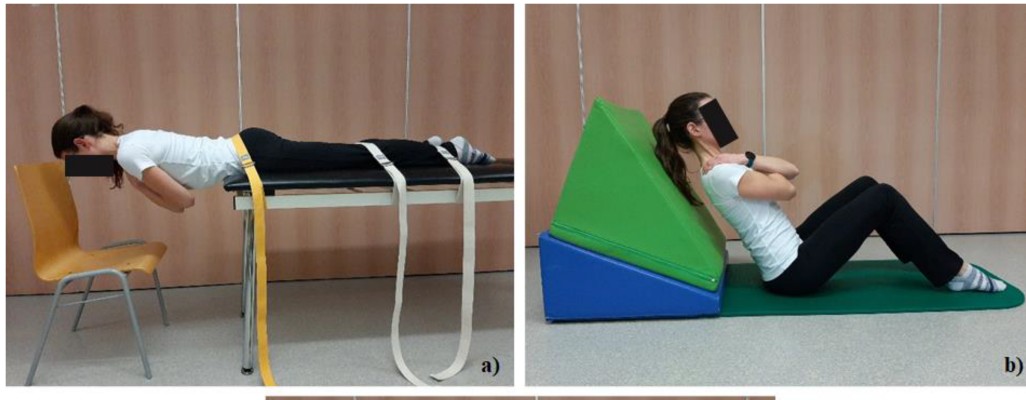

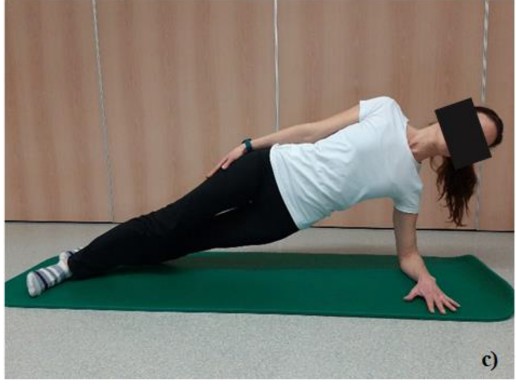

**Figure 1 Test positions in tests A.** (A) Trunk extensor endurance test (EX-A), (B) Trunk flexor endurance test (FL-A), and (C) the side bridge test (LM-A) for left side (L-LM-A).

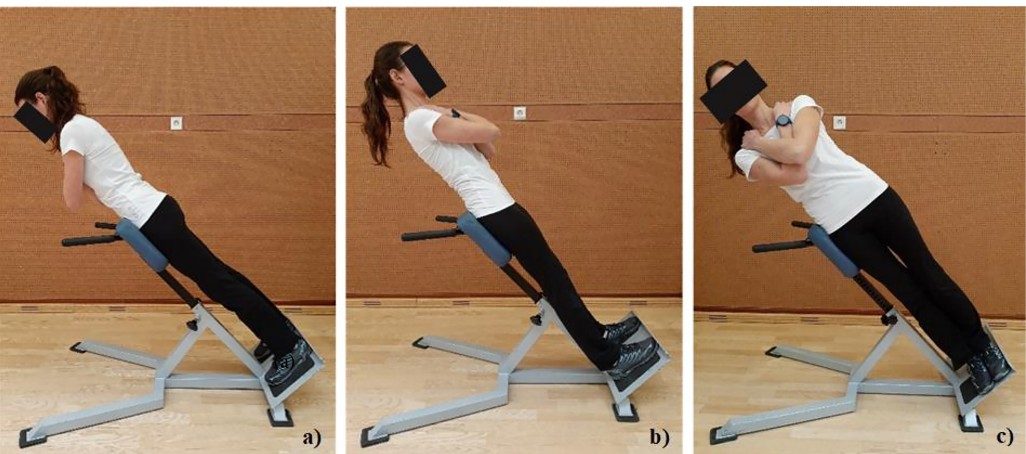

**Figure 2 Test positions in tests B.** (A) Trunk extensor isometric hold test (EX-B), (B) Trunk flexor isometric hold test (FL-B), and (C) Trunk lateral isometric hold test (LM-B) for left side (L-LM-B).

to the table using three straps. When ready, the participant held a horizontal body position as long as possible with their arms crossed over their chest (Fig. 1A). Unlike the original Biering-Sørensen test (*Biering-Sørensen, 1984*) we did not stop testing at 240s if participants could keep the test position longer.

- The starting position in the trunk flexor endurance test (Test FL-A) requires the participant to be seated on the floor, with their trunk supported at a 60° incline position, while keeping the head and neck in a neutral position. Knees and hips were flexed at 90° and arms crossed over the chest. The support of the trunk was then moved about 10 cm back, while the participant maintains the 60° position, parallel to distant support (Fig. 1B) (*McGill, Childs & Liebenson, 1999*).

- The side bridge test (Test LM-A) was performed in the side-lying position with hips and knees extended, aligning the top foot placed in front of the lower foot (heel-to-toe position). Participant placed the lower arm under the body with the elbow positioned directly under the shoulder and forearm placed palm down for balance and support. When ready, the participant placed the upper arm on the side of the body while lifting their hips off the mat and hold this position as long as possible (Fig. 1C). The same test position was used on both sides and was named according to the side of arm support.

B tests, the "alternative" group of trunk muscle endurance tests, consisted also of four different positions, all performed on a 45° Roman chair (Sokol Group, Sokolgym, Slovenia): trunk extensor isometric hold test, trunk flexor isometric hold test, and trunk lateral isometric hold test on both right and left sides. The height of the apparatus was set prior to the testing: when participant was in the prone position on an apparatus (in sports footwear), the apparatus support edge was aligned to the height of the individuals' SIAS line. Therefore, testing positions in B tests were (Fig. 2):

- The trunk extensor isometric hold test (Test EX-B) (*Ledoux, Dubois & Descarreaux, 2012*) was performed in the prone position with the anterior side of the pelvis supported on the apparatus. Arm support was allowed prior to the beginning of the test. The test started when participants crossed their arms over their chest and leaned their trunk forward in a line with their lower body, maintaining extended knees (Fig. 2A).

- For the trunk flexor isometric hold test (Test FL-B) the participant was supported in the supine position with the Roman chair set at 45° and the upper body in a straight position. Testing was initiated when the participant crossed their arms over their chest and leaned their trunk, neck and head back in line with their lower body while maintaining their knees in an extended position (Fig. 2B).

- The trunk lateral isometric hold test (Test LM-B) was performed in a lateral position for both left and right sides named according to the side in the up position (*Ledoux, Dubois & Descarreaux, 2012*; *Pagé & Descarreaux, 2012*). During the left lateral isometric hold test the participant was positioned on the right side on the Roman chair set at 45° and the trunk superior to the SIAS line unsupported (Fig. 2C). Participants were instructed to lean their trunk and head in line with their lower limbs which were placed on above the other, maintaining their knees in an extended position and their arms crossed over their chest.

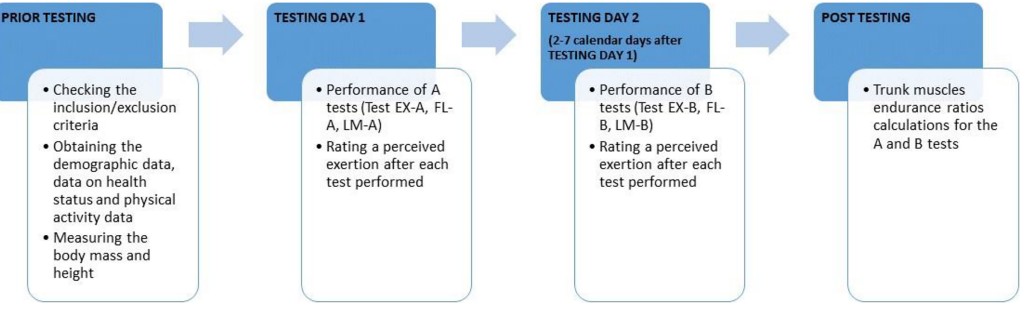

**Figure 3 Study procedures timeline.** EX, Trunk extensors endurance; FL, Trunk flexors endurance; LM, Lateral trunk muscles endurance.     

## Study procedures

Prior to the endurance tests, the body mass and height were measured by a stand scale with the height rod (7835; Soehnle, Backnang, Germany).

First, the participants performed A tests. B tests were performed within a week, but with at least 2 days' (48 h) time period in-between the muscle endurance test groups (Fig. 3). The objective of all tests was to hold an appropriate static position for as long as possible. Tests were terminated when there appeared a deviation from a neutral spine position, an increase in the low-back arch or when participant was no longer able to maintain the position. Time was measured by a stopwatch (BT-2136251; Basetech, Fürstenfeldbruck, Germany) and was recorded in record sheet in seconds.

Within each group of tests, a minimum of 5 min break was provided between the tests to facilitate recovery (*Ledoux, Dubois & Descarreaux, 2012*). The order of tests execution was randomly assigned to each participant (pre-prepared test forms), whereby a participant did perform A and B tests in the same order (*e.g.*, if participant performed A tests in sequence: (1) Test FL-A, (2) Test LM-A, and (3) Test EX-A, then the same order was also used with the B tests (Test FL-B—Test LM-B—Test EX-B) for this participant). Thus, a total of six different test sequences were used in the study, named by trunk muscle group assessed: (a) EX-FL-LM, (b) FL-LM-EX, (c) LM-EX-FL, (d) EX-LM-FL, (e) FL-EX-LM, and (f) LM-FL-EX. Total of 11 to 12 participants were assigned into each of six different test sequences listed above.

In the first minute after each test performed, participants were asked to rate a perceived exertion at the Borg Rating of Perceived Exertion (RPE) scale (*Borg, 1970*) (Fig. 3). For LM a perceived exertion was rated for left and right side together, as one rate of perceived exertion to perform LM test position. Rated perceived exertion (scale range: 6–20) was recorded in the record sheet for the corresponding test.

There were some confounding factors in this study that we have tried to minimize as much as possible. First, there was a possibility of delayed onset muscle soreness after the first group of tests implementation. Therefore, the second group of tests has always been performed by at least 48 h delay, as that kind of muscle response usually last for 24–48 h after the exercise (or testing in our case) (*Fleckenstein et al., 2021*). Second, the participants got the instructions to avoid high intensity physical activities and/or trainings or been

**Table 1 Classification of ratios between trunk muscles calculated as ratios of time (in seconds) achieved at each endurance test (summarized after McGill (2016)).**

| Ratio of comparison | Criteria for good ratio between muscles |
| --- | --- |
| FL (s): EX (s) | Ratio less than 1.0 |
| R-LM (s): L-LM (s) | Scores should be no greater than 0.05 from a balanced score 1.0 |
| L-LM (s): EX (s) | Ratio less than 0.75 |
| R-LM (s): EX (s) | Ratio less than 0.75 |

Note:
FL, Trunk flexors endurance; EX, Trunk extensors endurance; R-LM, Lateral trunk muscles (right side) endurance; L-LM, Lateral trunk muscles (left side) endurance.

awake late into the night a day before the testing, in order to assure the optimal re-testing conditions.

There were no familiarization sessions prior to testing. Testing procedures were explained and demonstrated by the researcher before each measurement and tests. Researcher who performed all the measurements and testing was a physiotherapist with a master's degree and 5 years of clinical experience.

### Trunk muscles endurance ratios calculation

The ratio between three trunk muscle groups—EX, FL and LM—were calculated for each group of tests performed. Four ratios of trunk muscles endurance comparison were calculated to determine if there is a good or poor ratio between muscles (McGill, 2016) (Table 1).

After ratio calculation described above, the comparison of division into good or poor ratio calculation has been made for the A tests and B tests in each participant.

### Study statistics

Data analysis was performed using an Excel program (Microsoft Corporation, Washington, USA), and IBM SPSS Statistics 26 (IBM, New York, USA). After checking for the normality of the distribution of variable, the measurements were statistically compared with the non-parametric Wilcoxon signed test for related-samples (Altman, 1991). Additionally, for the comparison of muscle endurance ratios between the two groups of tests (using the categories of "good" or "poor" ratios), McNemar's test was used (Altman, 1991). The statistical significance was set at $p < 0.05$ for all analyses.

### Ethical considerations

This study was approved by the National Medical Ethics Committee of the Republic of Slovenia (0120-220/2019/6). All participants were volunteers and gave their written informed consent prior the study.

## RESULTS

### Trunk muscle endurance ratios

The only statistically significant ($p < 0.05$) difference in trunk muscle endurance ratios between A and B tests was for FL:EX (Table 2).

**Table 2 Results in muscle endurance ratios calculated by holding time achieved at each endurance test, compared between both groups of tests ($n = 68$).**

| Endurance ratio | Group of tests | Median | IQR | $p$ | $z$ |
|---|---|---|---|---|---|
| FL:EX | A | 1.2 | 0.7–1.6 | <0.001 | −6.434 |
|  | B | 0.6 | 0.3–0.8 |  |  |
| R-LM:L-LM | A | 1.0 | 0.8–1.2 | 0.107 | −1.613 |
|  | B | 0.9 | 0.7–1.1 |  |  |
| L-LM:EX | A | 0.5 | 0.3–0.6 | 0.783 | −0.275 |
|  | B | 0.5 | 0.3–0.6 |  |  |
| R-LM:EX | A | 0.4 | 0.3–0.6 | 0.148 | −1.448 |
|  | B | 0.4 | 0.3–0.6 |  |  |

Note:
IQR, Interquartile range; p, Asymptotic sig. (2-sided test); z, test statistic (Wilcoxon signed ranks test).

**Table 3 Results in division consistency (in %) of calculated muscle endurance ratios into "good" or "poor" ratio between trunk muscles in both groups of tests ($n = 68$).**

| Endurance ratio | A = B (%) | (A = poor) ∩ (B = good) (%) | (A = good) ∩ (B = poor) (%) | $p$ |
|---|---|---|---|---|
| FL:EX | 60.3 | 39.7 | 0 | <0.001 |
| R-LM:L-LM | 70.6 | 17.6 | 11.8 | 0.503 |
| L-LM:EX | 83.8 | 10.3 | 5.9 | 0.549 |
| R-LM:EX | 82.3 | 11.8 | 5.9 | 0.338 |

Note:
A = B – share of unified division into "good" or "poor" ratio in both groups of tests, meaning: ((A = good) ∩ (B = good)) + ((A = poor) ∩ (B = poor)); (A = poor) ∩ (B = good) – share of reverse division in group A and B (resulted as "poor" ratio in tests A and as "good" ratio in tests B); (A = good) ∩ (B = poor) – share of reverse division in group A and B (resulted as "good" ratio in tests A and as "poor" ratio in tests B); p, Exact sig. (2-sided test) (McNemar test).

Furthermore, when the ratios of the A and B tests were compared, using the category "good" or "poor", as defined in Table 1, the test groups were unified in all three ratios except for FL:EX (Table 3).

### Endurance holding times

B tests revealed significantly longer holding times ($p < 0.001$) in the three out of the four tests as compared to A tests (Table 4).

### Rated perceived effort to perform tests

Results on perceived effort rated by participants when performed each test of A and B tests are shown in Table 5. As compared to tests A, tests B for extensors and lateral trunk muscles were rated as slightly easier to perform ($p_{RPE-EX} < 0.001$; $p_{RPE-LM} = 0.001$).

## DISCUSSION

The purpose of our study was to study the characteristics and usefulness of potentially better alternative test positions when measuring trunk muscle endurance and to verify if B tests shows comparable results to A tests in terms of muscle endurance ratio calculations, holding times and rated perceived effort to perform each test. Our study revealed that

**Table 4  Results in holding times (in seconds) comparison by each test for both groups of tests ($n = 68$).**

| Test | Group of tests | Median (s) | IQR | $p$ | $z$ |
|------|---------------|-----------|-----|-----|-----|
| EX | A | 125.5 | 104.8–182.8 | <0.001 | −7.168 |
| | B | 284.0 | 213.0–342.3 | | |
| FL | A | 149.0 | 91.3–247.0 | 0.824 | −0.222 |
| | B | 139.0 | 97.0–259.5 | | |
| L-LM | A | 61.0 | 48.3–80.8 | <0.001 | −7.149 |
| | B | 131.5 | 95.5–158.5 | | |
| R-LM | A | 63.5 | 45.8–77.3 | <0.001 | −7.122 |
| | B | 113.0 | 86.3–148.8 | | |

Note:
IQR, Interquartile range; p, Asymptotic sig. (2-sided test); z, test statistic (Wilcoxon signed ranks test).

**Table 5  Results in rated perceived effort (RPE) of tests performance comparison by each test for both groups of tests ($n = 68$).**

| Test | Group of tests | Median RPE rate | IQR | $p$ | $z$ |
|------|---------------|-----------------|-----|-----|-----|
| EX | A | 13.0 | 12.0–14.0 | <0.001 | −4.527 |
| | B | 11.0 | 10.0–13.0 | | |
| FL | A | 13.0 | 12.0–15.0 | 0.551 | −0.596 |
| | B | 13.0 | 12.0–15.8 | | |
| LM | A | 14.0 | 12.3–15.8 | 0.001 | −3.280 |
| | B | 13.0 | 12.0–15.0 | | |

Note:
RPE, Rated perceived effort; IQR, Interquartile range; p, Asymptotic sig. (2-sided test); z, test statistic (Wilcoxon signed ranks test).

trunk muscle endurance tests on the Roman chair (B tests) have comparable characteristics as already established tests (A tests) except for FL:EX ratio ($p < 0.001$). When holding time was considered, only the FL holding time was comparable between the tests with other holding times being longer in B tests ($p_{EX} < 0.001$; $p_{LM} < 0.001$). In terms of usefulness, all of the B tests showed to be more feasible, easier and quicker to implement by evaluator/researcher (no extra preparation needed prior the testing, as external fixation *etc.*) and in terms of the perceived exertion rated by participants, B tests are as slightly easier to perform as compared to the A tests. Therefore, the results of our study partially confirmed our main hypothesis that tests groups A and B shows comparable results in terms of the muscle endurance ratio calculations (three of four ratios), with B tests having longer holding times (three of four tests) and being slightly easier to perform (two of three tests).

There are several differences that need to be taken into account when considering these two groups of tests. First, the testing positions of the tests differ in terms of the angle of the participants' body in relation to the floor. For example, the angle in the Test EX-A is 0° (participants' body is parallel to the floor) while in the Test EX-B, the angle is 45°; the angle in the Test FL-A is 60°, while in the Test FL-B, the angle is 45°; in the Test LM-A, the angle is approximately 20°, while in Test LM-B, the angle is 45°. Different angles described above

cause a very different low back loading demands and torques that potentially affect the holding times in each testing position (*Tuff, Beach & Howarth, 2020*).

The hip and shoulder muscles are loaded differently between A and B tests. For hip muscles, this is reflected in FL endurance tests. In these two tests, the hip muscles can affect the results differently since in the Test FL-A, hips are in flexion, while in the Test FL-B, hips are in extension. The ratio FL:EX is the only one that is not comparable among A and B tests and the position of the hips during the tests might be the reason for that. The influence of hip muscles has been pointed out by other authors as well (*Pagé, Dubois & Descarreaux, 2011*; *Castro-Piñero et al., 2021*). *Pagé, Dubois & Descarreaux (2011)* when measured the activation of muscles with surface electromyography (EMG), reported significant earlier fatigue in lower limb muscles as compared to the trunk FL muscle. The difference in hip position in the A and B test, and consequently different activation of hip muscles during the A and B test, might be the reason for the differences in the holding times.

The activation of shoulder muscle was different in the tests for LM of the trunk. In the Test LM-A (Fig. 1C), shoulder muscles were activated during the test (*Juan-Recio et al., 2022*), while on the other hand, they were not activated in the Test LM-B (Fig. 2C). This might be the reason that the holding times are longer in the B tests.

When compared holding times by absolute times our study revealed significantly longer holding times in three of four B tests as compared to the A tests (EX, L-LM and R-LM; $p < 0.001$). For the trunk EX endurance participants have held the test position in Test EX-B more than twice as long as in Test EX-A. According to the test position, this difference is not surprising, since the angle of body inclination in the test position is different in the Test EX-B (Fig. 2A) than in Test EX-A (Fig. 1A). Similarly, holding times for LM were almost twice as long in the Test LM-B as compared to the Test LM-A. In addition, in the Test LM-B, shoulder joint was not stressed. *Ledoux, Dubois & Descarreaux (2012)* described the protocol of "lateral isometric hold test" (our Test LM-B) as an alternative to the side bridge test (our Test LM-A) when evaluate muscle endurance in older adults and adults with upper limb injuries or muscle weakness.

Although only healthy participants were included in our study, only one of 68 participants obtained a "good" ratio between right and left side of LM (*i.e.*, less than 5% difference between sides for ratio R-LM:L-LM) in both tests groups. Any greater difference in holding times suggest trunk muscle imbalance (or "poor ratio between muscle groups") according to *McGill (2016)*. Some authors have already indicated doubt as to whether such strict norms are clinically relevant in characterizing individuals with a history of disabling back problems or an increased risk of back problems (*Pagé & Descarreaux, 2012*).

Taking into account the results obtained in RPE of each test performance, we can summarize that B tests are easier in trunk EX and LM endurance testing as compared to the A tests, but similar in trunk FL endurance testing. Therefore, in terms of usefulness and feasibility in clinical setting assessing, trunk muscles endurance by B tests will be more appropriate for the elderly and weak individuals (*e.g.*, during rehabilitation) or patients with upper limb problems as already suggested by *Ledoux, Dubois & Descarreaux (2012)* for EX and LM. In addition, since the body position angle do not change in the B tests and

there is no need for additional stabilisation, B tests are easier to implement to any population.

The aim of our study was to study the characteristics and usefulness of potentially better alternative test positions (B tests) when measuring trunk muscle endurance. By the unified body position angle of inclination (all four muscle groups are tested in 45° angle of inclination), and unified position of pelvis and both lower limbs (therefore all four muscle groups are tested in open kinetic chain) B tests showed better characteristics when compared to the A tests. The testing protocols for trunk EX and LM used in group of B tests have already been described by other (*Ledoux, Dubois & Descarreaux, 2012*; *Pagé & Descarreaux, 2012*), while the testing protocol for trunk FL endurance (Test FL-B) has not been described and our study is the first which investigated this.

There were some limitations of our study. First, EMG activities of the muscles were not recorded and as described in the discussion knowing which muscle is activated and which muscle fatigue first, is important for interpreting the results. However, it was out of the scope of our study. In future studies, EMG recording of trunk and hip muscles should be encouraged in order to ensure that trunk muscles are the ones that activate and fatigue during the test. Studies on test characteristics (validity and reliability) and usefulness of trunk muscle endurance tests on the Roman chair in different population groups are needed. Next, we did not base the sample size on statistical power analysis. We could not do this because the necessary data did not exist in the literature. We tried to mitigate this shortcoming with a study design in which the same participants received experimental and control treatments, thus allowing a smaller sample (*Wang & Ji, 2020*). As there is smaller number of participants needed in this kind of the studies, we have assumed that 68 participants should be sufficient sample to assess the tests' differences. Additionally, our study can be seen as a pilot study that provided information on the effect size. This will enable the determination of the optimal sample in the future studies, with which we will be able to assess the comparability of the two test batteries even more reliably than it was possible in the present study. Finally, one could argue that the participants were not divided by gender. However, the study was not focused in determination of impact of different characteristics of study participants but on the differences between two sets of tests within participants. Definitely, it would be very interesting to study the possible gender differences when using the Roman chair apparatus for trunk muscle endurance assessment in the future.

## CONCLUSIONS

Trunk muscle endurance tests on the Roman chair (B tests) revealed some comparable characteristics and some potentially more feasible characteristics as already established tests (A tests). A and B tests were comparable in most of the calculated trunk muscle endurance ratios as well as in dividing into "good" or "poor" ratio between trunk muscles resulted from calculated ratios. Tests were not comparable in holding times and RPE ratios; there were longer holding times observed in three of four B tests *vs.* A tests, and for two of three assessed trunk muscle groups, B tests were also slightly easier to perform. The key difference obtained between test groups were differences in the testing positions

and thereby fatigue of other muscle groups (not only trunk muscles) observed especially in A tests. In terms of the characteristics and usefulness obtained in our study, B tests can be used as an alternative to standard tests, particularly since all four muscle groups are tested in the unified body position, angle of inclination and all in open kinetic chain. They are also easier to implement to different populations.

Results obtained in our study are important for the physiotherapy and public health professionals as LBP is still one of the leading functional disability cause among the musculoskeletal disorders. Therefore, feasible trunk muscle endurance assessment could present an effective evidence-based public health measure in musculoskeletal rehabilitation processes as also in its prevention. A proposed battery of tests could be very useful in continuous monitoring of the lower back musculature condition in children, adolescents, and emerging adults, especially students. However, as LBP increases with age for both genders, there is a need of effective trunk muscle endurance assessment and regular physical activity promotion on health maintenance or improvement in all population groups.

## ACKNOWLEDGEMENTS

We thank to all the participants for their cooperation.

### Funding

This work was supported by the Slovenian Research Agency (research core funding No. P3-0388 and research core funding No. P3-0429). The funders had no role in study design, data collection and analysis, decision to publish, or preparation of the manuscript.

### Grant Disclosures

The following grant information was disclosed by the authors:
Slovenian Research Agency: P3-0388 and P3-0429.

### Competing Interests

The authors declare that they have no competing interests.

### Author Contributions

- Maja Petrič conceived and designed the experiments, performed the experiments, analyzed the data, prepared figures and/or tables, authored or reviewed drafts of the article, and approved the final draft.
- Lijana Zaletel-Kragelj conceived and designed the experiments, analyzed the data, authored or reviewed drafts of the article, and approved the final draft.
- Renata Vauhnik conceived and designed the experiments, analyzed the data, authored or reviewed drafts of the article, and approved the final draft.

## Human Ethics

The following information was supplied relating to ethical approvals (*i.e.*, approving body and any reference numbers):

Study was approved by the National Medical Ethics Committee of the Republic of Slovenia (0120-220/2019/6).

## Data Availability

The raw data is available at: PETRIČ, Maja, ZALETEL-KRAGELJ, Lijana in VAUHNIK, Renata, 2022, Characteristics and usefulness of trunk muscle endurance tests on the Roman chair in healthy adults: supplemental data [na spletu]. 2022. [Dostopano 13 november 2022]. Pridobljeno https://repozitorij.uni-lj.si/IzpisGradiva.php?lang=slv&id=142472.

## Supplemental Information

Supplemental information for this article can be found online at http://dx.doi.org/10.7717/peerj.14469#supplemental-information.

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
