# Peer review of "Characteristics and usefulness of trunk muscle endurance tests on the Roman chair in healthy adults"

_PeerJ, doi:10.7717/peerj.14469_

## Round 0.1 · original submission · Major Revisions

The reviewers support the idea that the article has merit. however, improvements must be performed. Please carefully consider the comments and suggestions of the three reviewers.

·

Basic reporting

Dear authors,

The article is valuable in terms of its subject and scope. Thank you for the effort you put into your research.

After a few minor corrections, I think your research is fit to be published for peerj.

-Please review some English mistakes.
-Please include your main hypothesis in your research at the end of the introduction.

Experimental design

I think that the experimental design and method section are sufficient.

Validity of the findings

I think the results section is sufficient.

Additional comments

After the minor verifications I mentioned in the Basic reports, your research is suitable for publication.

·

Basic reporting

General comments:
Title
Are presented satisfactorily.
Abstract
I consider the abstract to be relatively well written, however, it would be necessary to insert some absolute values, in addition to the statistical values. I also suggest that the last part suggesting other studies be taken from the abstract and that the methodology be better detailed.
Introduction
The introduction is not starting from general to specific. It should initially present a more general approach and gradually address the problem (gap) and then present the objective.
The problem must be better identified. The introduction should focus on the proposed test construct.
Please make a link between the problem and the proposed objective.
It would be indicated that the study hypotheses to be answered were presented.

Experimental design

Methods
It should present more clearly the design of the study. A CONSORT or time line, should be presented in order to get a better view of the study design.
The sample should be better explained with the number of subjects presented initially and then present the inclusion and exclusion criteria. Which program or reference was used to arrive at the number of participants. Some statistical program was used. Why this number of evaluated.
The methodology is confusing, it suggests that it be divided into topics, design, sample, instruments, procedures and statistics.
The place of insertion of figures and tables must be pointed out in the text.
The figures must meet ethical standards and a stripe must be placed as a way of preserving the identity of the participants.
It was mentioned that there would be a five-minute, 48-hour break for recovery. Please put a reference so that this rest time is enough for the proposed.
The instruments and procedures must be referenced from other studies.
If a new instrument was proposed, there should be a topic regarding the construct of this instrument to justify its use.
Statistical treatment should be better detailed in order to better follow what has been done. I suggest reviewing the design and statistical analysis for possible validation of said test and also consulting Cohen (1988).

Validity of the findings

As attached notes

Additional comments

Results
As mentioned earlier, as the design and statistics are not properly described, the results end up being poor. To use a new instrument, some parameters must be respected. I suggest that the results be reviewed in view of the mentioned in the methodology.
Discussion
Despite the discussion being relatively well written, the aforementioned problems do not allow the results presented to be enough to give an answer and credibility to the evaluations.
I suggest that the methodology be adequate and from there that the studies presented in the discussion are used, solidifying the answer.
Conclusion
In view of what was mentioned in the methodology, the conclusion ends up being weakened. It would also be important to focus more on the practical applications of the findings.
References
Please review the formatting of references. Of the 24 references, 11 are current and 13 are more than five years old. please update.

Reviewer 3 ·

Basic reporting

See "additional comments"

Experimental design

See "additional comments"

Validity of the findings

See "additional comments"

Additional comments

Thank you for the opportunity to review this article. The paper addresses a novel under-researched area. However, there are some questions that need to be addressed to the manuscript.

Specific comments are provided below:

TITLE
Add the type of population of your study

INTRODUCTION
Why are higher in women respect to men? Explain it. (line 48)
Please add a short introduction (2-3 phrases) before the aims where you explain the reasons of this study. (line 92)

METHODS
Include more information about the study design (i.e. longitudinal or transversal...) (line 103)
Have you include patients with chronic pain LBP or previous serious injuries in the back? (line 105-110)
Explain more about these questionnaires (line 114-115)
Can you show the reliability data of these tests? (line 153-155)
How did you calculate the body mass and height? Explain it (line 179)
Include reliability data (ICC and CV) and the stopwatch model. (line 185)
How long? Specify it. (line 196)
Were there familiarization sessions prior to testing? (line 202)
Why do not you divide by gender? It would be very interesting to observe possible differences by gender (line 214)
Include normality test. (line 214)

RESULTS
Include this information in "study participants, recruiting and inclusion criteria" In addition, add information on body composition divided by gender. (line 228)
Add more information about statistical analysis (relevant content) in “endurance holding times” and “rated perceived effort to perform tests” sections. (line 240 and 244).

DISCUSSION
Provide information about your statistical results in the discussion section (line 248)
Add the aim/s of your study in this first paragraph (line 249)
And the consequences are... (line 263)
Can you compare your results with those of other studies? (line 301-307)
This information is not relevant in a discussion section. If you want you can include it in the methods section (line 309-315)
This information is relevant in the introduction and conclusions sections. Please, change it. (line 327-338).

---

## Round 0.2 · Minor Revisions

The authors improved the article in the latest stage of revision. However, we would like to ask for additional changes in methods and results to enhance the clarity of information and guarantee replicability.

·

Basic reporting

Congratulations

Experimental design

Congratulations

Validity of the findings

Congratulations

Additional comments

Congratulations

·

Basic reporting

General comments:
Title
Are presented satisfactorily.
Abstract
It is written in a structured way, however, the methodology is written in a very summarized way which ends up making the findings and conclusions of the article.
Please confirm that the Keywords are listed as descriptors in health sciences.
Introduction
It should initially present a more general approach and gradually address the problem (gap) and then present the objective.
Mentioning that there are few studies does not seem to me to be a good problem. Although the introduction is well written, better identification of the problem would help a lot in sustaining the objectives.
Methods
It should present more clearly the design of the study. A CONSORT or time line, should be presented in order to get a better view of the study design.
The sample should be better explained with the number of subjects presented initially and then present the inclusion and exclusion criteria.
Statistical treatment should be better detailed in order to better follow what has been done. Please consult Cohen (1988), data such as effect size would help a lot to have a better view of the findings.
Results
Are presented satisfactorily. However, the mentioned in the methodology must be observed and the main results must be better explained.
Discussion
Are presented satisfactorily.
Conclusion
Are presented satisfactorily.
References
We ask you to confirm the formatting of the references. Of the 27 references, 13 are current and 14 have been published for more than five years. Please update references.
Overview
The manuscript presented addresses a relevant research topic.
It would be advisable to do a general review.

Experimental design

Methods
It should present more clearly the design of the study. A CONSORT or time line, should be presented in order to get a better view of the study design.
The sample should be better explained with the number of subjects presented initially and then present the inclusion and exclusion criteria.
Statistical treatment should be better detailed in order to better follow what has been done. Please consult Cohen (1988), data such as effect size would help a lot to have a better view of the findings.

Validity of the findings

Results
Are presented satisfactorily. However, the mentioned in the methodology must be observed and the main results must be better explained.

Additional comments

As attached notes

Reviewer 3 ·

Basic reporting

No comment

Experimental design

No comment

Validity of the findings

No comment

Additional comments

Congratulations on a much improved manuscript.

---

## Author Rebuttal · Round 0.2

University of Ljubljana
Faculty of Health Sciences
Zdravstvena pot 5
1000 Ljubljana
Slovenia

Ljubljana, 4.9.2022

Dear Prof. Filipe Manuel Clemente,

We would like to thank you and your team for comments and suggestions with reference to our manuscript. We have addressed these comments and suggestions below and marked the changes in the manuscript. The reviewers' comments are in blue and italics, while our response is in bold.

Please let us know if you require any additional information.

Kind regards,

Dr. Renata Vauhnik

On the behalf of all authors

*Reviewer 1 (Ali Kerim Yılmaz)*
*Basic reporting*
*Dear authors,*
*The article is valuable in terms of its subject and scope. Thank you for the effort you put into your research. After a few minor corrections, I think your research is fit to be published for PeerJ. Please review some English mistakes.*

**AUTHORS' RESPONSE: Thank you for your comment. The manuscript has been additionally checked by an English native speaker.**

*Please include your main hypothesis in your research at the end of the introduction.*

**AUTHORS' RESPONSE: Thank you for your suggestion. The main hypothesis has been added at the end of the introduction (line 119-121):** "We hypothesized that tests groups will show comparable results in terms of the muscle endurance ratios in healthy adults, with tests on the Roman chair having longer holding times; and being easier to perform."

*Experimental design*
*I think that the experimental design and method section are sufficient.*
*Validity of the findings*
*I think the results section is sufficient.*
*Additional comments*
*After the minor verifications I mentioned in the Basic reports, your research is suitable for publication.*
* * *
*Reviewer 2 (Felipe Aidar)*
*Basic reporting*
*General comments:*
*Title*
*Are presented satisfactorily.*
*Abstract*
*I consider the abstract to be relatively well written, however, it would be necessary to insert some absolute values, in addition to the statistical values. I also suggest that the last part suggesting other studies be taken from the abstract and that the methodology be better detailed.*

**AUTHORS' RESPONSE: Thank you for your comment and your suggestion. Absolute values have been added to the Abstract (line 35-42) and the last**

**sentence has been taken from the Abstract. In addition, as suggested we have added some methodology data (line 31-33):** "Results of each participant were compared for trunk muscle endurance ratio calculations, holding times and rated perceived effort for A and B tests."

*Introduction*
*The introduction is not starting from general to specific. It should initially present a more general approach and gradually address the problem (gap) and then present the objective.*
*The problem must be better identified. The introduction should focus on the proposed test construct.*

**AUTHORS' RESPONSE: Thank you for your comment and your suggestion. The Introduction has been revised/corrected (line 75-81):** "Therefore, feasible trunk muscle endurance assessment could present an effective evidence-based public health measure for prevention and treatment in musculoskeletal rehabilitation of lumbar spine conditions. A proposed battery of tests could be very useful in continuous monitoring of the lower back musculature in children, adolescents, and emerging adults, especially students. In these population groups, according to the recent research, which showed alarming results (Jurak et al., 2021; Horvat Tišlar, Starc & Kukec, 2022), such monitoring will be more than necessary" **Line 95-99**: "As currently available and used tests have some limitations in the use of different population groups and positions used for testing, there is a need for optimizing the trunk muscle endurance assessment protocols. One, potentially better, alternative protocol when testing trunk muscle endurance is testing it on the 45° Roman chair apparatus (Ledoux, Dubois & Descarreaux, 2012; Pagé & Descarreaux, 2012)."

*Please make a link between the problem and the proposed objective.*

**AUTHORS' RESPONSE: Thank you for your suggestion. A link between the problem and the proposed objective has been added (line 110-113): "**Although some of the trunk muscle endurance tests on the Roman chair have been described in a few studies (for extensors and lateral trunk muscles endurance testing), the characteristics and usefulness of the proposed group of tests (i. e. for each of three major trunk muscle groups) have not been studied."

*It would be indicated that the study hypotheses to be answered were presented.*

**AUTHORS' RESPONSE: The main hypothesis has been added at the end of the introduction (line 119-121):** "We hypothesized that tests groups will show comparable results in terms of the muscle endurance ratios in healthy adults, with tests on the Roman chair having longer holding times; and being easier to perform."

*Experimental design*
*Methods*
*It should present more clearly the design of the study. A CONSORT or time line, should be presented in order to get a better view of the study design.*

**AUTHORS' RESPONSE: Thank you for your comment. We have added this information to the manuscript (line 125-127):** "The study was designed as a quasi-experimental repeated-measures design (Rogers & Révész, 2019). There was a single group of participants who took part in two different trunk muscle endurance testing as described below."

*The sample should be better explained with the number of subjects presented initially and then present the inclusion and exclusion criteria. Which program or reference was used to arrive at the number of participants. Some statistical program was used. Why this number of evaluated.*

**AUTHORS' RESPONSE: Thank you for your comment. We have added the number of the subjects as you have suggested before the inclusion and exclusion criteria (line 130-132):** "Sixty-eight healthy adults (51 women and 17 men; age: 31.9±7.2 years; body height: 1.7±0.1 m; body mass: 66.4±12.1 kg; body mass index: 23.0±3.5 kg/m2) participated in the study and all participants finished the study." **We did not perform statistical power analysis to define the sample size, since our research is using a new approach of measuring/testing trunk muscle endurance and there are no studies available with appropriate data to calculate the sample size. However, since our study was designed as a quasi-experimental repeated-measures design there was a single group of participants who took part in both trunk muscle endurance test methods. This allows us to achieve a very good control of possible biased factors such as gender, age, health status, etc. The more detailed information regarding this have been added to the manuscript (lines 125-127):** "The study was designed as a quasi-experimental repeated-measures design (Rogers & Révész, 2019). There was a single group of participants who took part in two different trunk muscle endurance testing as described below." **And added in the Discussion section (lines 378-384):** "Next, we did not base the sample size on statistical

power analysis. We could not do this because the necessary data did not exist in the literature. We tried to mitigate this shortcoming with a study design in which the same participants received experimental and control treatments, thus allowing a smaller sample (*Wang & Ji, 2020*). In addition, our study can also be regarded as a pilot study in which we obtained information about the effect size, which can allow determination of optimal sample size in future studies."

*The methodology is confusing, it suggests that it be divided into topics, design, sample, instruments, procedures and statistics.*

**AUTHORS' RESPONSE: Thank you for your comment. We have followed your suggestion and divided the method section into the topics:** Study design, Study sample, Study instruments, Study procedures and Study statistics**.**

*The place of insertion of figures and tables must be pointed out in the text.*

**AUTHORS' RESPONSE: We have added the place of insertion of figures and tables in the manuscript.**

*The figures must meet ethical standards and a stripe must be placed as a way of preserving the identity of the participants.*

**AUTHORS' RESPONSE: The stripes have been added to all figures.**

*It was mentioned that there would be a five-minute, 48-hour break for recovery. Please put a reference so that this rest time is enough for the proposed.*

**AUTHORS' RESPONSE: Thank you for your comment. We have added the reference in the Methods section (line 219):** "(Ledoux, Dubois & Descarreaux, 2012)."

*The instruments and procedures must be referenced from other studies.*
*If a new instrument was proposed, there should be a topic regarding the construct of this instrument to justify its use.*

**AUTHORS' RESPONSE: Thank you for your comment. In our study a new approach for measuring/testing trunk muscle endurance has been proposed and a rationale for using this new approach is described in the Introduction section from lines 83-113.**

*Statistical treatment should be better detailed in order to better follow what has been done. I suggest reviewing the design and statistical analysis for possible validation of said test and also consulting Cohen (1988).*

**AUTHORS' RESPONSE: Thank you for your comment. We did not perform statistical power analysis to define the sample size, since our research is using a new approach of measuring/testing trunk muscle endurance and there are no studies available with appropriate data to calculate the sample size. However, since our study was designed as a quasi-experimental repeated-measures design there was a single group of participants who took part in both trunk muscle endurance test methods. This allows us to achieve a very good control of possible biased factors such as gender, age, health status, etc. The more detailed information regarding this have been added to the manuscript (lines 125-127):** "The study was designed as a quasi-experimental repeated-measures design (Rogers & Révész, 2019). There was a single group of participants who took part in two different trunk muscle endurance testing as described below." **And please see Lines 258-262:** "After checking for the normality of the distribution of variable, the measurements were statistically compared with the non-parametric Wilcoxon signed test for related-samples (*Altman, 1991*). Additionally, for the comparison of muscle endurance ratios between the two groups of tests (using the categories of "good" or "poor" ratios), McNemar's test was used (*Altman, 1991*)."

*Additional comments*
*Results*
*As mentioned earlier, as the design and statistics are not properly described, the results end up being poor. To use a new instrument, some parameters must be respected. I suggest that the results be reviewed in view of the mentioned in the methodology.*

**AUTHORS' RESPONSE: We hope that with our explanation regarding the study design and statistical analysis written above, we have answered to your doubt about our methodology and that the results section does not require changes.**

*Discussion*
*Despite the discussion being relatively well written, the aforementioned problems do not allow the results presented to be enough to give an answer and credibility to the evaluations.*
*I suggest that the methodology be adequate and from there that the studies presented in the discussion are used, solidifying the answer.*

**AUTHORS' RESPONSE: Please see our answers above for the methodology. We hope our answers for methodology are satisfactory and that the discussion is satisfactory in the present form.**

*Conclusion*
*In view of what was mentioned in the methodology, the conclusion ends up being weakened. It would also be important to focus more on the practical applications of the findings.*

**AUTHORS' RESPONSE: Thank you for your comment. We have added the practical applications of the findings to the manuscript (line 404-412):** "Results obtained in our study are important for the physiotherapy and public health professionals as LBP is still one of the leading functional disability cause among the musculoskeletal disorders. Therefore, feasible trunk muscle endurance assessment could present an effective evidence-based public health measure in musculoskeletal rehabilitation processes as also in its prevention. A proposed battery of tests could be very useful in continuous monitoring of the lower back musculature condition in children, adolescents, and emerging adults, especially students. However, as LBP increases with age for both genders, there is a need of effective trunk muscle endurance assessment and regular physical activity promotion on health maintenance or improvement in all population groups."

*References*
*Please review the formatting of references. Of the 24 references, 11 are current and 13 are more than five years old. Please update.*

**AUTHORS' RESPONSE: Thank you for your comment. We have reviewed the formatting of references and corrected when required. To our knowledge there is only a small amount of relevant current studies in the research area addressed in this study and this is the reason for some references being older.**
* * *
*Reviewer 3 (Anonymous)*
*Basic reporting*
*See "additional comments"*
*Experimental design*
*See "additional comments"*
*Validity of the findings*
*See "additional comments"*
*Additional comments*

*Thank you for the opportunity to review this article. The paper addresses a novel under-researched area. However, there are some questions that need to be addressed to the manuscript.*

*Specific comments are provided below:*

*TITLE*
*Add the type of population of your study*

**AUTHORS' RESPONSE: Thank you for your comments. The type of population of our study has been added to the title (line 2):** "in healthy adults"
*INTRODUCTION*
*Why are higher in women respect to men? Explain it. (line 48)*

**AUTHORS' RESPONSE: The following has been added in the manuscript to clarify this (line 52-55):** "Low back pain is present in all age groups and the prevalence, incidence and YLDs rates are higher in women (**especially in the middle aged**) than in men and it increases with age for both genders (Hartvigsen et al., 2018; Chen et al., 2021; IHME, 2022)."
**And we don't think it is known why LBP is more problematic for women.**

*Please add a short introduction (2-3 phrases) before the aims where you explain the reasons of this study. (line 92)*

**AUTHORS' RESPONSE: Thank you for your comment. We have added the following to the manuscript (lines 95-99):** "As currently available and used tests have some limitations in the use of different population groups as well as in the positions used for testing, there is a need for optimizing the trunk muscle endurance assessment protocols. One, potentially better, alternative protocol when testing trunk muscle endurance is testing it on the 45° Roman chair apparatus (Ledoux, Dubois & Descarreaux, 2012; Pagé & Descarreaux, 2012)."
**And Lines 110-113: "**Although some of the trunk muscle endurance tests on the Roman chair have been described in a few studies (for extensors and lateral trunk muscles endurance testing), the characteristics and usefulness of the proposed group of tests (i. e. for each of three major trunk muscle groups) have not been studied."

*METHODS*
*Include more information about the study design (i.e. longitudinal or transversal...) (line 103)*

**AUTHORS' RESPONSE: The study was designed as a quasi-experimental repeated-measures design. The information has been added to the manuscript (lines 125-127):** "The study was designed as a quasi-experimental repeated-measures design (Rogers & Révész, 2019). There was a single group of participants who took part in two different trunk muscle endurance testing as described below."

*Have you include patients with chronic pain LBP or previous serious injuries in the back? (line 105-110)*

**AUTHORS' RESPONSE: We have included only healthy adults without any LBP or back injuries.**

*Explain more about these questionnaires (line 114-115)*

**AUTHORS' RESPONSE: Explanation has been added to the manuscript (line 141-143):** "Questions on health status and physical activity were intended to reveal if some exclusion criteria or contraindication exists among the candidates to participate in the study.*"*

*Can you show the reliability data of these tests? (line 153-155)*

**AUTHORS' RESPONSE: We were not able to find any reliability data of trunk muscle endurance tests performed on the Roman chair in the literature. Reliability testing would have been an obvious choice as to the usefulness of the Roman chair.**

*How did you calculate the body mass and height? Explain it (line 179)*
**AUTHORS' RESPONSE: We have measured the body mass and height by a stand scale with the height rod. The scale model was added in the manuscript (Soehnle, 7835, Germany) (line 210).**

*Include reliability data (ICC and CV) and the stopwatch model. (line 185)*

**AUTHORS' RESPONSE: The model of the stopwatch was added in the manuscript (Basetech, BT-2136251, Germany) (line 216). We were not able to find any reliability data on this stopwatch model.**

*How long? Specify it. (line 196)*

**AUTHORS' RESPONSE: Perceived exertion was rated in the first minute after each test was performed. The information has been added to the manuscript (line 228).**

*Were there familiarization sessions prior to testing? (line 202)*

**AUTHORS' RESPONSE: Familiarization sessions prior to the testing were not performed. We have added this information also to the manuscript (line 242):** "There were no familiarization sessions prior to testing."

*Why do not you divide by gender? It would be very interesting to observe possible differences by gender (line 214)*

**AUTHORS' RESPONSE: Thank you for your comment. We agree that it would be interesting to observe possible differences by gender. However, since our objective was to study the characteristics and usefulness of potentially better alternative test positions when measuring trunk muscle endurance, we had a single group of participants who took part in both two different trunk muscle endurance testing and this allows us to achieve a very good control of possible biased factors such as gender, age, health status, etc. We have included this suggestion and explanation to the Discussion section (lines 384-388):** "Finally, one could argue that the participants were not divided by gender. However, the study was not focused in determination of impact of different characteristics of study participants on the differences between two sets of tests within participants. Definitely, it would be very interesting to study the possible gender differences when using the Roman chair apparatus for trunk muscle endurance assessment in the future."

*Include normality test. (line 214)*

**AUTHORS' RESPONSE: Thank you for your comment. Normality of the variable values distribution was checked with the Shapiro Wilk's test. We have added**

**the information to the manuscript (lines 258-260):** "After checking for the normality of the distribution of variable values, the measurements were statistically compared with the non-parametric Wilcoxon signed test for related-samples (*Altman, 1991*)."

*RESULTS*
*Include this information in "study participants, recruiting and inclusion criteria"*
*In addition, add information on body composition divided by gender. (line 228)*

**AUTHORS' RESPONSE: We have included this information to the Methods section (lines 130-132):** "Sixty-eight healthy adults (51 women and 17 men; age: 31.9±7.2 years; body height: 1.7±0.1 m; body mass: 66.4±12.1 kg; body mass index: 23.0±3.5 kg/m2) participated in the study and all participants finished the study." **The body composition divided by gender was not added, since we had a single group of participants who took part in both two different trunk muscle endurance testing and this allows us to achieve a very good control of possible biased factors such as gender, age, health status, etc.**

*Add more information about statistical analysis (relevant content) in "endurance holding times" and "rated perceived effort to perform tests" sections. (line 240 and 244).*

**AUTHORS' RESPONSE: Thank you for your comment. More information have been added into the manuscript (lines 282-283):** "B tests revealed significantly longer holding times (p < 0.001) in the three out of four tests as compared to A tests (Table 4)." **And Lines 287-289:** "Results on perceived effort rated by participants when performed each test of A and B tests are shown in Table 5. As compared to tests A, tests B for extensors and lateral trunk muscles were rated as slightly easier to perform ($p_{RPE-EX}$ < 0.001; $p_{RPE-LM}$ = 0.001)."

*DISCUSSION*
*Provide information about your statistical results in the discussion section (line 248)*

**AUTHORS' RESPONSE: Thank you for your comment. We have added this information to the manuscript (lines 298-300, 335).**

*Add the aim/s of your study in this first paragraph (line 249)*

**AUTHORS' RESPONSE: Thank you for your comment. We have added this information to the manuscript (lines 293-296):** "The purpose of our study was to study the characteristics and usefulness of potentially better alternative test positions when measuring trunk muscle endurance and to verify if B tests shows comparable results to A tests in terms of muscle endurance ratio calculations, holding times and rated perceived effort to perform each test."

*And the consequences are... (line 263)*

**AUTHORS' RESPONSE: Thank you for your comment. We have added this information to the manuscript (lines 313-315):** "Different angles described above cause a very different low back loading demands and torques **that potentially affect the holding times in each testing position** (*Tuff, Beach & Howarth, 2020*)."

*Can you compare your results with those of other studies? (line 301-307)*

**AUTHORS' RESPONSE: Thank you for your comment. We have added this information to the manuscript (lines 355-358):** "Therefore, in terms of usefulness and feasibility in clinical setting assessing, trunk muscles endurance by B tests will be more appropriate for the elderly and weak individuals (e.g. during rehabilitation) or patients with upper limb problems **as already suggested by *Ledoux, Dubois & Descarreaux (2012)* for EX and LM**."

*This information is not relevant in a discussion section.*
*If you want you can include it in the methods section (line 309-315)*

**AUTHORS' RESPONSE: Thank you for your comment. We have included it in the Methods section of the manuscript (lines 234-240).**

*This information is relevant in the introduction and conclusions sections. Please, change it. (line 327-338).*

**AUTHORS' RESPONSE: Thank you for your comment. We have corrected and included in the Introduction and Conclusions sections of the manuscript (lines 75-81 and 404-412).**

---

## Round 0.3 · accepted · Accept

The article was improved during the revision stages and can be accepted in its current form.

·

Basic reporting

Revisions were made to the manuscript and the authors' responses. In view of the adjustments presented, I consider the manuscript in a condition to be published.

Experimental design

.

Validity of the findings

.

Additional comments

.